# Erythrodermic Psoriasis in the Context of Emerging Triggers: Insights into Dupilumab-Associated and COVID-19-Induced Psoriatic Disease

**DOI:** 10.3390/dermatopathology12020017

**Published:** 2025-06-09

**Authors:** Aya Fadel, Jayakumar Nithura, Zahraa F. Saadoon, Lamia Naseer, Angelo Lopez-Lacayo, Ligia Elena Rojas Solano, Chaveli Palau Morales, Robert J. Hernandez, Hussain Hussain

**Affiliations:** 1Department of Internal medicine, HCA Florida Kendall Hospital, Miami, FL 33175, USA; nithura2022@gmail.com (J.N.); zahraa.fadhil0178@gmail.com (Z.F.S.); lamianaseer9@gmail.com (L.N.); angelo.lopezlacayo@hcahealthcare.com (A.L.-L.); ligia.elers21@hotmail.com (L.E.R.S.); cp2121@mynsu.nova.edu (C.P.M.); 2Department of Internal Medicine, Hackensack Meridian Hospital, Ocean University Medical Center, Brick, NJ 08724, USA; 3Department of Infection Disease, HCA Florida Kendall Hospital, Miami, FL 33175, USA; roberthernandezmd@gmail.com; 4Department of Obstetrics, Gynecology and Reproductive Sciences, University of Miami Miller School of Medicine, Miami, FL 33136, USA

**Keywords:** psoriasis, COVID-19, SARS-CoV-2, T-cell, erythrodermic psoriasis, IL-4, dupilumab, monoclonal antibodies

## Abstract

Psoriasis is a chronic immune-mediated inflammatory skin disorder characterized by keratinocyte hyperproliferation, impaired epidermal barrier function, and immune dysregulation. The Th17/IL-23 axis plays a central role in its pathogenesis, promoting the production of key pro-inflammatory cytokines such as IL-17, IL-23, and TNF-α, which sustain chronic inflammation and epidermal remodeling. Emerging evidence suggests that SARS-CoV-2 may trigger new-onset or exacerbate existing psoriasis, likely through viral protein-induced activation of toll-like receptors (TLR2 and TLR4). This leads to NF-κB activation, cytokine release, and enhanced Th17 responses, disrupting immune homeostasis. Erythrodermic psoriasis (EP), a rare and severe variant, presents with generalized erythema and desquamation, often accompanied by systemic complications, including infection, electrolyte imbalance, and hemodynamic instability. In a murine model of SARS-CoV-2 infection, we found notable cutaneous changes: dermal collagen deposition, hair follicle destruction, and subcutaneous adipose loss. Parallel findings were seen in a rare clinical case (only the third reported case) of EP in a patient with refractory psoriasis, who developed erythroderma after off-label initiation of dupilumab therapy. The patient’s histopathology closely mirrored the changes seen in the SARS-CoV-2 model. Histological evaluations also reveal similarities between psoriasis flare-ups following dupilumab treatment and cutaneous manifestations of COVID-19, suggesting a shared inflammatory pathway, potentially mediated by heightened type 1 and type 17 responses. This overlap raises the possibility of a latent connection between SARS-CoV-2 infection and increased psoriasis severity. Since the introduction of COVID-19 vaccines, sporadic cases of EP have been reported post-vaccination. Although rare, these events imply that vaccine-induced immune modulation may influence psoriasis activity. Our findings highlight a convergence of inflammatory mediators—including IL-1, IL-6, IL-17, TNF-α, TLRs, and NF-κB—across three triggers: SARS-CoV-2, vaccination, and dupilumab. Further mechanistic studies are essential to clarify these relationships and guide management in complex psoriasis cases.

## 1. Introduction

Psoriasis is a chronic immune-mediated inflammatory skin disorder marked by the hyperproliferation of keratinocytes and widespread systemic immune involvement. It affects approximately 7.5 million individuals in the United States, representing about 2–3% of the adult population [1]. The condition can manifest at any age, although it exhibits bimodal peak incidence—initially during adolescence and early adulthood, and later between the ages of 50 and 60 years [2]. Among its clinical subtypes, plaque psoriasis is the most prevalent, accounting for 80–90% of all cases [3]. The disease imposes a substantial burden on quality of life due to its visible lesions, intense pruritus, and association with numerous comorbidities, including psoriatic arthritis, metabolic syndrome, cardiovascular disease, and mood disorders such as depression [4]. Notably, the therapeutic targeting of the interleukin-23 (IL-23)/T-helper 17 (Th17) axis has demonstrated favorable clinical outcomes, particularly when individualized and timely interventions are implemented [5].

The underlying pathophysiology of psoriasis is multifactorial, involving a complex interplay between genetic susceptibility and immune dysregulation. Key susceptibility loci, notably the human leukocyte antigen (HLA-Cw6) allele, have been implicated in disease development [6]. Immunologically, psoriasis is driven by the activation of Th17 and Th1 cells, which secrete cytokines such as IL-17A, IL-22, and TNF-α. These cytokines stimulate keratinocyte proliferation, resulting in epidermal hyperplasia and the accelerated turnover of skin cells [7]. Additional immune cells—including dendritic cells, neutrophils, and macrophages—further propagate inflammation. Compounding this immune dysregulation, impaired skin barrier function due to reduced filaggrin expression renders the epidermis more vulnerable to external triggers such as infections, trauma, and psychological stress [8]. During disease flare-ups, this dysregulated immune activity intensifies, amplifying cytokine cascades, cellular infiltration, and tissue damage, culminating in the development of thickened, erythematous plaques [9].

Environmental triggers are critical modulators of disease activity, with infectious agents—particularly Streptococcus pyogenes and, more recently, SARS-CoV-2—being recognized as potent exacerbating factors [10]. Viral infections can initiate or aggravate psoriatic inflammation through several overlapping mechanisms. In the case of COVID-19, multiple case reports and mechanistic studies have observed the onset or worsening of psoriasis following SARS-CoV-2 infection [11]. Although the precise pathogenesis remains to be fully delineated, current evidence suggests that both the intact virus and its structural proteins—especially the spike (S) and nucleocapsid (N) proteins—may initiate or exacerbate autoimmunity in genetically predisposed individuals [12].

These viral components are known to interact with pattern-recognition receptors (PRRs), including Toll-like receptors (TLRs), on both immune and epithelial cells. Engagement of TLR2, TLR3, and TLR7 activates downstream signaling pathways, particularly nuclear factor-kappa B (NF-κB) and interferon regulatory factors (IRFs), leading to the increased production of pro-inflammatory mediators such as IL-6, TNF-α, and type I interferons [13]. This pro-inflammatory milieu promotes the recruitment and activation of autoreactive T-cells, fostering the chronic inflammatory environment characteristics of psoriatic pathology [14]. Moreover, molecular mimicry between SARS-CoV-2 antigens and host proteins may induce a breakdown in immune tolerance, triggering autoreactive responses. Additional mechanisms, such as bystander T-cell activation, epitope spreading, and the dysfunction of regulatory T-cells (Tregs), further amplify the autoimmune cascade [15]. Collectively, these findings underscore the potential of SARS-CoV-2 to act as a powerful immunological trigger in the initiation or exacerbation of psoriasis, particularly in individuals with pre-existing genetic and immunological susceptibility.

Erythrodermic psoriasis is a rare but severe and potentially life-threatening variant of psoriasis, characterized by widespread erythema and desquamation, affecting more than 75–90% of the body’s surface area. Unlike localized plaque psoriasis, erythrodermic psoriasis disrupts thermoregulation and skin barrier function, often leading to systemic symptoms such as fever, chills, dehydration, tachycardia, and even high-output cardiac failure. This condition may arise de novo or as an exacerbation of existing plaque psoriasis, frequently triggered by factors such as infection, the abrupt withdrawal of systemic corticosteroids, stress, or certain medications. Due to its extensive cutaneous involvement and systemic manifestations, erythrodermic psoriasis often necessitates hospitalization and requires prompt multidisciplinary intervention to prevent complications such as sepsis, electrolyte imbalance, and organ failure. Its pathophysiology is thought to involve a hyperactivation of the Th17/IL-23 axis and an amplified cytokine storm, contributing to both cutaneous inflammation and systemic immune dysregulation.

## 2. Case Study

A 70-year-old male with a medical history of hypertension, chronic alcohol use, and a 20-year history of plaque psoriasis presented to the emergency department with a rapidly worsening, intensely pruritic, and erythematous rash diffusely involving his entire body (Figure 1). The patient had no known drug allergies and was recently initiated on dupilumab 300 mg subcutaneously for refractory psoriasis. This treatment was prescribed off-label by his dermatologist, as dupilumab is not approved for psoriasis management in the United States. Prior to this, multiple therapeutic options had been attempted without significant improvement. The patient received his second dose four days before admission. Shortly thereafter, he developed progressive erythema accompanied by severe pruritus and burning sensations, initially localized to the upper back and subsequently spreading to involve most of his body.

Upon presentation, he was afebrile with a blood pressure of 180/83 mmHg, pulse of 100 bpm, respiratory rate of 21 breaths per minute, and an oxygen saturation of 100% on room air. He was alert and oriented, without signs of acute cardiopulmonary or neurologic compromise. Dermatologic examination revealed generalized erythema and diffuse scaling with areas of epidermal denudation over the extremities, trunk, and face (Figure 1). Re-epithelialization was beginning in some regions. No mucosal, genital, or ocular involvement was observed. The affected skin was markedly tender to palpation, and Nikolsky’s sign was negative.

Initial laboratory investigations revealed normocytic anemia, leukocytosis with a marked left shift, and neutrophilia. The basic metabolic panel, liver enzymes, thyroid function tests, and urinalysis were all within normal limits. Toxicology screening was negative. Given the extensive skin involvement and leukocytosis accompanied by systemic signs, sepsis was suspected. A Code Sepsis was called, prompting immediate blood cultures, intravenous fluid resuscitation, and empirical broad-spectrum antibiotics, later de-escalated upon negative cultures and clinical improvement.

A skin biopsy was obtained from an area of active erythema. Figure 2: Histopathological findings disclosing hematoxylin and eosin (H&E)-stained skin biopsy reveal a marked thickening of the dermal layer accompanied by intercellular fluid accumulation. Notable histopathological changes were observed within the dermis, including a dense infiltration of inflammatory cells and the presence of interstitial bullae. Additionally, extensive collagen deposition was evident, as indicated by the red arrow, suggesting active fibrotic remodeling. Figure 3: The yellow star highlights a region of dense inflammatory cell infiltration within the dermal layer. These immune cells are surrounded by extensive collagen deposition, likely mediated by profibrotic signaling pathways involving transforming growth factor-beta (TGF-β) and other collagen-inducing cytokines (IL-4, 13, and TNF). This pattern suggests an active inflammatory-fibrotic process consistent with severe psoriatic dermopathy. Figure 4: The biopsy reveals marked pathological changes in the dermis, including dense inflammatory cell infiltration, tissue congestion, extensive collagen deposition consistent with fibrotic remodeling, and interstitial fluid accumulation leading to bullae-like formations. These findings collectively suggest a severe inflammatory and fibrotic process contributing to the disruption of dermal architecture.

The patient was managed conservatively with supportive care, including topical emollients, systemic corticosteroids, antihypertensives, intravenous hydration, and antihistamines such as diphenhydramine. Throughout hospitalization, his cutaneous symptoms gradually improved, with a marked resolution of erythema and discomfort by day three. By day seven, the patient was stable and discharged home with instructions to permanently discontinue dupilumab and to follow up closely with the Dermatology department for further management of his psoriasis. This case emphasizes a rare but severe paradoxical reaction to dupilumab therapy, manifesting as erythrodermic psoriasis, in a patient with long-standing psoriatic disease. Early recognition and multidisciplinary management are crucial for preventing life-threatening complications.

## 3. SARS-CoV-2 and Psoriasis

In the context of the COVID-19 pandemic, multiple case reports (Table 1) and mechanistic studies have suggested a potential relationship between SARS-CoV-2 infection and the triggering or exacerbation of psoriasis [3]. Understanding the pathophysiological mechanisms underlying these conditions is critical for advancing both treatment strategies and patient care. Recent evidence suggests that COVID-19 may exacerbate or even induce psoriasis in previously unaffected individuals. The proposed mechanisms include viral-triggered immune dysregulation, particularly via the activation of macrophages, dendritic cells, and Toll-like receptors (TLRs). This leads to a surge in pro-inflammatory cytokines such as interleukin (IL)-6, IL-1β, and tumor necrosis factor-alpha (TNF-α), which is known to be cytokine storm. Furthermore, the activation of Toll-like receptors (TLRs), particularly TLR3 and TLR7, by viral components can further enhance type I interferon responses, indirectly promoting the IL-23/Th17 axis, a key pathway in psoriasis development. This immune cascade may initiate new-onset psoriasis or exacerbate existing disease in genetically predisposed individuals [11].

Clinical presentations of COVID-19-associated psoriasis are varied, including guttate, erythrodermic, and pustular forms [23,24]. Notably, some cases have occurred in individuals without a prior history of psoriasis, indicating that SARS-CoV-2 infection may unmask latent predispositions [25]. Additionally, treatments administered during COVID-19, such as hydroxychloroquine and systemic corticosteroids, have been implicated in psoriasis flare-ups. For instance, hydroxychloroquine exacerbates psoriasis, possibly due to its effects on plasmacytoid dendritic cells and T-cell activation [25]. The cessation of immunosuppressive therapies during infection may also contribute to disease exacerbation.

The psychological impact of the COVID-19 pandemic, including stress, anxiety, and social isolation, has been significant and may contribute to psoriasis flare-ups [26]. Stress activates the hypothalamic–pituitary–adrenal (HPA) axis, leading to increased cortisol levels, which can disrupting immune regulation and skin barrier function [26]. Furthermore, lifestyle changes during the pandemic, such as reduced physical activity, altered sleep patterns, and dietary changes, may negatively affect psoriasis management [26]. The interruption of regular healthcare services and challenges in accessing treatments have further compounded these issues, emphasizing the need for comprehensive care approaches during such global health crises [26].

We examined the impact of Coronavirus infection on multiple organ systems—including the liver, brain, kidney, and skin—using a previously established animal model [27,28,29,30,31,32]. As part of this investigation, we conducted a comprehensive dermatological analysis to characterize both acute and long-term cutaneous manifestations of COVID-19 [27]. In the acute phase, we observed significant structural alterations in the skin, including disruption of the epidermal layer, an increased number of hair follicles, extensive collagen deposition within the dermis, and marked hyperplasia of the sebaceous glands. Additionally, there was a noticeable thinning of the panniculus carnosus and adventitial layer—findings that align closely with the cutaneous pathology reported in human studies [27]. In contrast, the long-COVID phase was characterized by more severe and irreversible changes [27]. Histological analysis revealed a complete absence of hair follicles in both the epidermal and dermal layers, extensive destruction of adipose tissue, and profound degeneration of the epidermal layer [27]. Furthermore, we also confirmed the presence of viral particles in long-term infection [27].

The histological analysis revealed marked differences between control and long-term SARS-CoV-2-infected mice. Panel A (control, Figure 5) shows a well-preserved hypodermis, intact adipose tissue, organized hair follicles, and normal epidermal and interstitial structures. In contrast, Panel B (12-month post-infection, Figure 5) displays fibrotic remodeling, subcutaneous fat loss, epidermal thinning, and dense collagen deposition with inflammatory infiltration, indicating chronic cutaneous injury and long-term structural damage from viral infection.

Shortly after viral entry, Toll-like receptor 2 (TLR2) is rapidly activated, triggering a downstream signaling cascade that robustly stimulates the nuclear factor-kappa B (NF-κB) pathway [33]. This molecular activation leads to the upregulation of a wide array of pro-inflammatory cytokines, including interleukins (IL-1, IL-6, IL-17, and IL-12), tumor necrosis factor-alpha (TNF-α), and chemokines such as chemokine ligand-2 (CCL2) [34,35,36]. In parallel, additional inflammatory mediators, such as interferons and histamine, are released, intensifying the recruitment of immune cells—particularly dendritic cells, macrophages, and T lymphocytes—to the affected tissues [35]. Although this immune activation is intended to contain viral propagation, the resultant excessive infiltration of inflammatory cells into the skin can paradoxically contribute to collateral tissue damage. In the epidermis, this manifests as the disruption of keratinocyte homeostasis, the degradation of structural proteins, and the loss of barrier function. These pathogenic features closely mirror the immunopathological mechanisms observed in psoriasis.

In psoriasis, the persistent activation of TLRs, particularly TLR2 and TLR4, along with sustained NF-κB signaling, drives chronic inflammation and promotes the differentiation and expansion of pathogenic Th1 and Th17 cells [37,38]. This leads to the continuous production of IL-17, IL-23, and TNF-α—cytokines that are central to psoriatic plaque formation and keratinocyte hyperproliferation [37,38]. Thus, the exaggerated inflammatory response triggered by viral infections such as SARS-CoV-2 may serve as both an initiator and amplifier of psoriatic disease, particularly in genetically susceptible individuals [27]. The overlap in immunological pathways emphasizes a possible mechanistic link between viral infections and the onset or exacerbation of psoriasis.

## 4. Psoriasis/Erythrodermic Psoriasis and COVID Vaccines

Since the introduction of COVID-19 vaccines, there have been occasional reports of erythrodermic psoriasis following vaccination; see Table 2. These events are rare, and the causal relationship remains unclear, although vaccine-induced immune activation is a proposed mechanism. COVID-19 vaccines, particularly mRNA-based types, can transiently modulate immune responses, potentially influencing underlying autoimmune or inflammatory conditions. For individuals with a history of psoriasis, especially those with severe or unstable disease, clinicians should monitor for potential flare-ups post-vaccination and consider appropriate preemptive or therapeutic strategies if needed.

## 5. Erythrodermic Psoriasis

Our patient developed severe EP following Dupilumab therapy, accompanied by systemic complication. The presence of rapid collagen deposition highlights the potential of this medication to trigger significant pathological changes in the skin. Its pathophysiology represents an exacerbation of the immune dysregulation and keratinocyte dysfunction characteristic of plaque psoriasis, amplified by systemic inflammatory and vascular responses. Genetic susceptibility is strongly associated with HLA-Cw6 and other genes such as IL-23R, TNFAIP3, and CARD14, which regulate immune activity [39,40,45,46]. Mutations or polymorphisms in these genes contribute to overactive inflammatory pathways and reduced immune tolerance, predisposing individuals to severe disease manifestations [45]. Immune dysregulation is central to EP (Figure 6), involving both the innate and adaptive immune systems. Dendritic cells, upon activation, secrete pro-inflammatory cytokines like IL-12 and IL-23, which polarize T-cells toward Th1 and Th17 phenotypes [47,48,49]. Th17 cells secrete IL-17, IL-22, and IL-21, promoting keratinocyte activation, proliferation, and cytokine release. Concurrently, Th1 cells produce interferon-gamma (IFN-γ) and tumor necrosis factor-alpha (TNF-α), further amplifying the inflammatory cascade [47,48,49]. Regulatory T-cells (Tregs), which ordinarily modulate immune responses, are dysfunctional in EP, leading to loss of immune tolerance and unchecked inflammation [47,48,49]. In acute-on-chronic psoriasis, transforming growth factor-beta (TGF-β) plays a critical role in promoting dermal fibrosis by stimulating fibroblast activation and collagen synthesis [47]. Persistent TGF-β signaling leads to excessive extracellular matrix deposition, contributing to tissue remodeling and thickening of the dermis. This fibrotic response may exacerbate chronic inflammation and impair normal skin architecture in severe or long-standing psoriatic lesions.

Keratinocyte hyperproliferation and dysfunction are major drivers of EP [44]. Cytokines released by immune cells stimulate keratinocyte proliferation, resulting in the impaired differentiation and accelerated turnover of skin cells. In turn, activated keratinocytes secrete cytokines like IL-1β, IL-6, and TNF-α, as well as chemokines such as CXCL1 and CXCL8, which recruit more immune cells to the skin, sustaining the inflammatory loop [50,51,52,53,54]. The molecular pathways involved include the IL-23/IL-17 axis, which drives Th17-mediated responses; TNF-α signaling, which activates NF-κB and inflammatory gene expression; and Janus kinases, signal transducers, and activators of transcription (JAK-STAT) pathways, stimulated by cytokines like IL-6 and IL-22, which promote keratinocyte proliferation while inhibiting differentiation [50,51,52,53,54]. Additionally, IL-36, a member of the IL-1 family, is overexpressed in EP and further amplifies inflammation by activating dendritic cells and neutrophils [50,51,52,53,54].

Systemic inflammation is a hallmark of EP and distinguishes it from other forms of psoriasis [55]. Cytokines such as IL-17 and TNF-α increase vascular permeability, leading to erythema, edema, and systemic heat loss [56]. The widespread involvement of the skin results in a compromised barrier, facilitating protein and fluid loss that can cause hypoalbuminemia and metabolic disturbances [56]. Elevated levels of acute-phase reactants like C-reactive protein (CRP) reflect the systemic nature of the disease, and the catabolic state induced by widespread inflammation exacerbates these metabolic challenges. Barrier dysfunction plays a critical role in EP [56]. The impaired production of structural proteins such as filaggrin and loricrin compromises the skin’s integrity, making it susceptible to microbial invasion and environmental triggers, which exacerbate inflammation. Triggers such as infections (e.g., Staphylococcus aureus or streptococcal infections), psychological stress, physical trauma (the Koebner phenomenon), or abrupt withdrawal of systemic treatments, including corticosteroids, can precipitate or worsen EP [56].

Dupilumab’s suppression of Th2 pathways may unmask underlying EP in predisposed individuals by allowing for the Th1 and Th17 responses to dominate the immune landscape [57]. This cytokine imbalance, characterized by decreased IL-4 and IL-13 signaling, removes the regulatory effects these cytokines exert on the Th1 and Th17 pathways, leading to the unchecked production of psoriatic cytokines such as IL-17, IL-23, and TNF-α [57]. Additionally, IL-4 and IL-13 influence keratinocyte behavior, and their inhibition may disrupt keratinocyte homeostasis, further exacerbating the hyperproliferation and impaired differentiation characteristics of psoriatic skin changes [57]. Together, these mechanisms contribute to the development or exacerbation of psoriasis, including its severe erythrodermic form, in certain patients receiving Dupilumab therapy.

We found both serological and histopathological similarities among three potential psoriasis-exacerbating triggers—dupilumab treatment, SARS-CoV-2 infection, and COVID-19 vaccination—suggesting the involvement of a shared inflammatory pathway. Key immunological mediators, including interleukins (IL-1, IL-6, IL-17), TNF-α, TLRs, and NF-κB, play pivotal roles in the pathogenesis of psoriasis by driving pro-inflammatory cascades and activating T lymphocytes. These pathways appear to be commonly engaged across the three triggers. Although direct mechanistic studies linking COVID-19 vaccination to psoriasis flare-ups are still lacking, accumulating clinical evidence supports the hypothesis that COVID-19 vaccines may represent novel immunological stimuli capable of initiating or aggravating psoriatic disease.

Moreover, histopathological evaluations have revealed striking similarities between the cutaneous changes observed in psoriasis flare-ups following dupilumab administration and those noted in COVID-19-related skin manifestations. These findings raise the possibility of a shared inflammatory pathway, potentially mediated by heightened type 1 and type 17 immune responses. This overlap may point to a long-term or latent connection between coronavirus infection and increased incidence or severity of psoriasis. Further mechanistic and longitudinal studies are needed to clarify these associations and assess whether SARS-CoV-2 or its immune response modifiers contribute to chronic skin inflammation in predisposed individuals.

## 6. Treatment Strategies

The treatment of erythrodermic psoriasis remains challenging and typically requires a multidisciplinary approach. Its initial management often involves hospitalization and supportive care to address fluid resuscitation, electrolyte imbalances, and prevent secondary infections. Systemic therapies, including methotrexate, cyclosporine, and acitretin, have traditionally been used to control acute flare-ups [1]. More recently, biologic agents targeting specific inflammatory pathways have shown promise in improving outcomes for patients with EP. TNF inhibitors, such as infliximab and adalimumab, as well as IL-17 inhibitors like secukinumab, have demonstrated efficacy in controlling the severe inflammation characteristics of this condition [1]. However, despite these advances, challenges remain in achieving long-term remission, and the response to treatment can be variable.

For mild-to-moderate psoriasis, topical agents remain the first line of treatment. These include corticosteroids, vitamin D analogs (e.g., calcipotriol), tazarotene, and calcineurin inhibitors [1,58]. In EP, topical therapies are often adjunctive due to the severity of systemic symptoms. Narrowband UVB phototherapy is effective for moderate-to-severe plaque psoriasis. However, its use in EP is limited because of the risk of burning on widespread inflamed skin [1,58]. Traditional systemic therapies include methotrexate, cyclosporine, and acitretin. Methotrexate inhibits dihydrofolate reductase and reduces T-cell proliferation. Cyclosporine suppresses calcineurin-mediated T-cell activation, and acitretin, a retinoid, normalizes keratinocyte differentiation [58]. Cyclosporine is particularly favored for acute EP due to its rapid onset of action [58].

Biologics have revolutionized psoriasis management. Agents targeting TNF-α (e.g., etanercept, adalimumab, infliximab), IL-12/23 (ustekinumab), IL-17 (secukinumab, ixekizumab), and IL-23 (guselkumab, tildrakizumab) are highly effective and generally well tolerated [1,58]. In EP, infliximab and ustekinumab have shown particular efficacy in rapidly reducing systemic inflammation [58]. During the COVID-19 pandemic, the continuation of biologic therapy has been a topic of debate. Data suggest that patients on biologic agents do not necessarily experience worse COVID-19 outcomes. The abrupt discontinuation of biologic agents may in fact contribute to psoriatic flare-ups.

## 7. Anticipated Treatments

Emerging therapies include Janus kinase (JAK) inhibitors, such as tofacitinib and deucravacitinib, which inhibit intracellular signaling of cytokine receptors involved in psoriasis [58,59]. TYK2 inhibitors, like deucravacitinib, offer a more targeted approach with fewer systemic side effects [60]. Additionally, research is exploring the modulation of the gut–skin axis, as gut microbiota has been implicated in the immune homeostasis relevant to psoriasis [61]. Probiotics, dietary interventions, and fecal microbiota transplantation are under investigation. Gene-editing technologies like CRISPR/Cas9 are being studied for their potential to correct psoriasis-associated genetic defects [62]. Nanoparticle-based drug delivery systems aim to enhance the local delivery of immunosuppressants while minimizing systemic toxicity [62]. In the post-COVID era, understanding how viral infections influence autoimmune and inflammatory pathways may lead to novel therapeutic targets and biomarkers predictive of disease onset or flare-ups in at-risk populations.

## 8. Conclusions

Psoriasis and its severe variant, erythrodermic psoriasis, represent complex immune-mediated disorders with significant morbidity. EP is characterized by a cascade of immune dysregulation, keratinocyte hyperactivity, and systemic inflammation. The interplay between genetic susceptibility, immune activation, and environmental triggers leads to widespread skin and systemic involvement, with significant morbidity and mortality risks. The COVID-19 pandemic has introduced new insights into how viral infections may trigger or exacerbate psoriasis, emphasizing the need for individualized and adaptive treatment strategies. Advances in biologic therapies, small-molecule inhibitors, and immunogenomics offer promising avenues for future intervention. Ongoing research into the interplay between infection, immunity, and genetics will continue to refine our understanding and management of this multifaceted disease.

## Figures and Tables

**Figure 1 dermatopathology-12-00017-f001:**
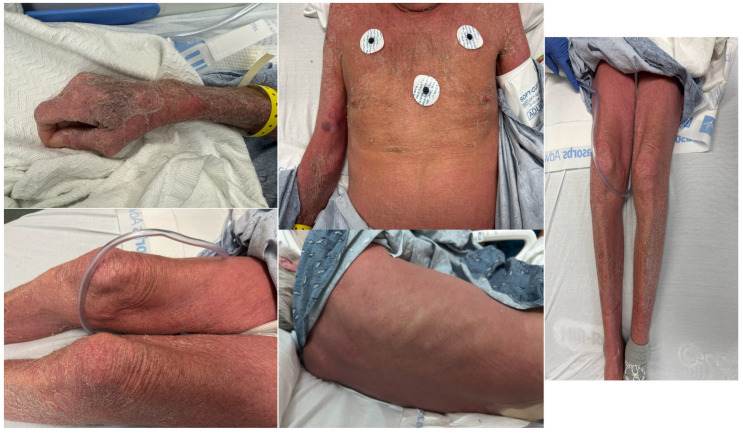
Diffuse erythema and desquamation were observed, accompanied by features of age-related skin changes, all consistent with a diagnosis of severe cutaneous erythrodermic psoriasis.

**Figure 2 dermatopathology-12-00017-f002:**
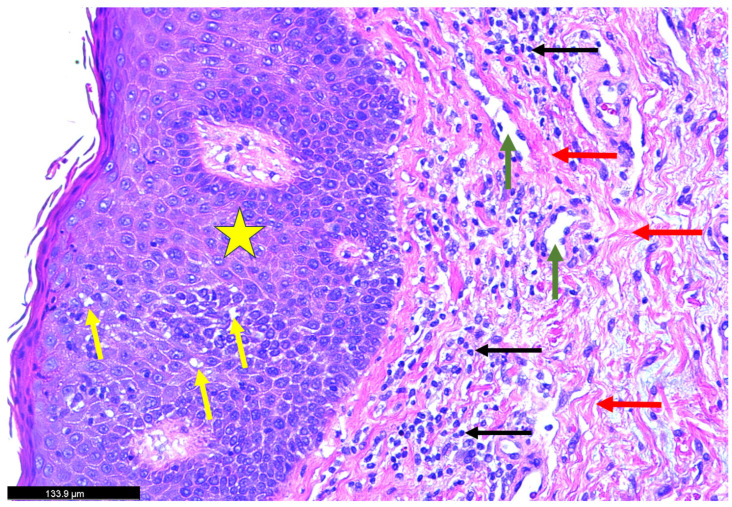
Histopathological findings of the skin biopsy (H\&E stain). The skin biopsy reveals marked thickening of the epidermal layer (yellow star), accompanied by notable intercellular fluid accumulation (yellow arrows). Dense inflammatory cell infiltration is observed (black arrows), along with increased collagen deposition around the capillaries (green arrows). Extensive collagen deposition within the dermis indicates ongoing fibrotic remodeling and tissue injury (red arrows).

**Figure 3 dermatopathology-12-00017-f003:**
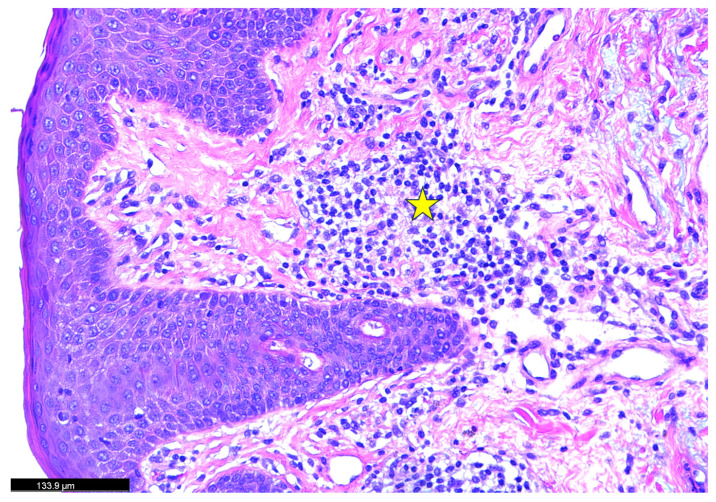
Histopathological findings of the skin biopsy (H&E stain). The yellow star indicates a region of intense inflammatory cell infiltration within the dermal layer. These inflammatory foci are surrounded by varying degrees of collagen deposition, suggestive of ongoing dermal remodeling and fibrotic response.

**Figure 4 dermatopathology-12-00017-f004:**
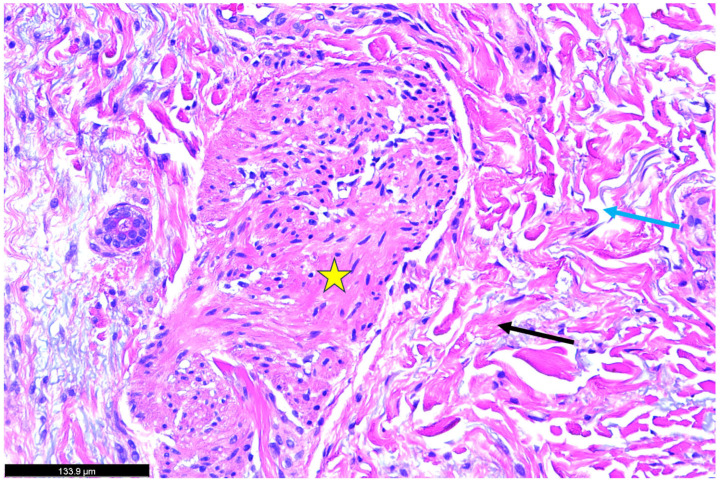
Histopathological findings of the skin biopsy (H&E stain). The yellow star indicates a region of intense inflammatory cell infiltration within the dermal layer and contains various inflammatory cells and tissue congestion. The yellow star indicates the smooth muscle with various cellular deposits, mild congestion, and collagen accumulation. The black arrow shows the nest of collage deposition, whereas the blue arrow reveals the interstitial fluid deposition and formation of bullae like pictures within the dermal layer.

**Figure 5 dermatopathology-12-00017-f005:**
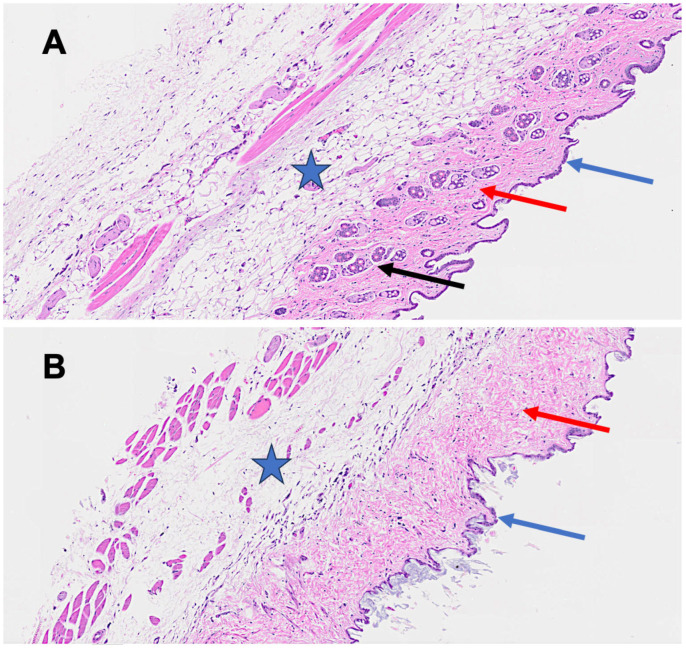
Mice histopathological findings of the skin biopsy (H&E stain). Panel (**A**): Control group (non-infected mice). The blue star highlights the normal hypodermis, while the black arrow indicates intact hair follicles. The red arrow denotes normal interstitial supportive tissues and cellular components. The blue arrow marks the normal well-organized epidermal layer. Panel (**B**): Long-term (12-month) SARS-CoV-2-infected mice. The star identifies a distorted hypodermis exhibiting fibrotic changes and the shrinkage of subcutaneous adipose tissue. The blue arrow points to the marked distortion and thinning of the epidermal layer. The red arrow highlights significant collagen deposition and inflammatory cell infiltration, with the severe disruption of normal dermal architecture.

**Figure 6 dermatopathology-12-00017-f006:**
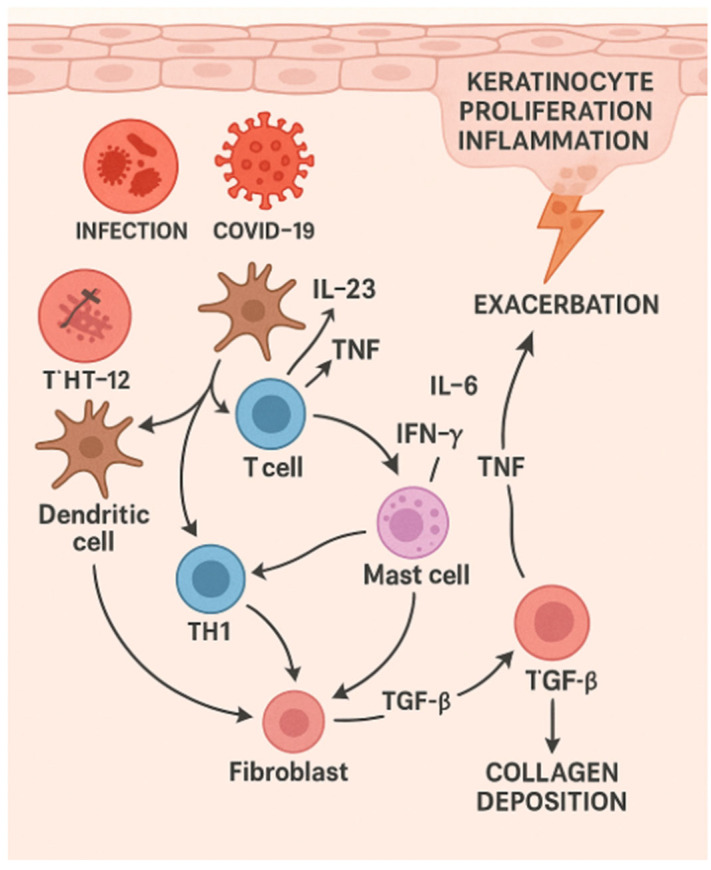
Various triggers (e.g., COVID-19 infection and vaccines, or other infections) initiate downstream signaling through dendritic cells, leading to the release of cytokines such as IL-23 and TNF, which stimulate T-cells. Activated T helper 1 (Th1) cells then produce pro-inflammatory cytokines like IFN-γ and TNF, further amplifying the immune response. Mast cells contribute additional cytokines, including IL-6 and TNF, reinforcing the inflammatory cascade. This cascade results in keratinocyte proliferation and inflammation in the epidermis, causing psoriatic plaque formation and the exacerbation of symptoms. Moreover, acute-on-chronic activation of fibroblasts by TGF-β promotes collagen deposition in the dermis, potentially contributing to tissue remodeling or fibrosis in chronic cases.

**Table 1 dermatopathology-12-00017-t001:** Disclosed various patients infected with SARS-CoV-2 related to different psoriasis.

Authors	Sex	Age	Type of Psoriasis	Concomitant Conditions and Medications	History of Psoriasis
Kutlu et al. [16]	Female	71	Psoriasis vulgaris	Oseltamivir and hydroxychloroquine	Yes
Agharbi et al. [17]	Male	55	Erythrodermic psoriasis	Dermocorticoids	Yes
Batubara et al. [18]	Male	46	Erythrodermic psoriasis	Narrowband ultraviolet B phototherapy	Yes
Martora et al. [19]	Male	61	Psoriasis exacerbation	Topical calcipotriol and betamethasone	Yes
Behrangi et al. [20]	2 cases: Male and Female	Male 50, Female 57	Male and Female: Erythrodermic psoriasis	Male: None Female: Diabetes, hepatic cirrhosis, insulin, methotrexate	Yes
Demiri et al. [21]	Male	60	Erythrodermic psoriasis	None	Yes
Janodia et al. [22]	Female	55	Guttate psoriasis	None	Yes

**Table 2 dermatopathology-12-00017-t002:** Presentations of new-onset or exacerbated psoriasis following COVID-19 vaccination.

Authors	Sex	Age	Type of Psoriasis	Concomitant Conditions and Medications	History of Psoriasis
Trepanowski N et al. [39]	Male	53	Erythrodermic psoriasis	None	Yes
Karampinis E et al. [40]	Male	56	Plaque psoriasis exacerbation	Dyslipidemia and diabetes mellitus	No
Nila AM et al. [41]	Male	58	Erythrodermic psoriasis	Topical corticosteroids	Yes
Tran TB et al. [42]	Female	30	Erythrodermic psoriasis	None	No
Lopez ED et al. [43]	Male	58	Erythrodermic psoriasis	Hypertension, psoriasis, prior intravenous drug use with heroine, L4-5 methicillin-resistant *Staphylococcus aureus* osteomyelitis, untreated hepatitis C, and tobacco	Yes
Onsun N et al. [44]	Male	72	Erythrodermic psoriasis	Hypertension and psoriasis	Yes

## Data Availability

A request from the corresponding author is required to obtain any data.

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
