# Peer review of "Erythrodermic Psoriasis in the Context of Emerging Triggers: Insights into Dupilumab-Associated and COVID-19-Induced Psoriatic Disease"

_dermatopathology, 2025, doi:10.3390/dermatopathology12020017_

Round 1

Reviewer 1 Report

Comments and Suggestions for Authors

Overall Assessment

This manuscript reports a case of dupilumab-induced erythrodermic psoriasis, provides a review of erythrodermic psoriasis triggered by COVID-19, and presents histopathological findings from a mouse model of long COVID. While each topic may be of interest individually, the rationale for combining these distinct subjects into a single manuscript is unclear. The lack of coherence among the topics makes the manuscript confusing. The authors should clearly explain why these three topics were integrated into one paper.

Concerning Dupilumab and Erythrodermic Psoriasis

The case of erythrodermic psoriasis following dupilumab administration raises several clinical concerns. In the reviewer's country, dupilumab is approved for atopic dermatitis, asthma, prurigo nodularis, and chronic spontaneous urticaria, but not for psoriasis.
- Was the patient treated off-label for psoriasis, or did they have a coexisting condition such as atopic dermatitis or asthma that justified the use of dupilumab?
- Is dupilumab approved for psoriasis in the United States or in the authors’ country? This point should be clarified.

Omission of COVID-19 Vaccination-Related Exacerbation

There are numerous reports of psoriasis flares and erythrodermic psoriasis following COVID-19 vaccination, not only infection.
- The manuscript does not address vaccination at all, which is a significant oversight.
- Did the patient in the reported case receive a COVID-19 vaccine? If so, the timing and type of vaccination should be discussed in relation to the onset of symptoms.

Unclear Purpose of Long COVID Mouse Model Pathology

The inclusion of histopathological findings from a long COVID mouse model is confusing.
- The relevance of these findings to erythrodermic psoriasis or the case presented is not explained.
- Moreover, the pathological features shown do not resemble those of erythrodermic psoriasis. If there is any similarity to human long COVID tissue, the authors should elaborate.

Technical Issue with Histological Image Presentation

- The histological images are presented at an oblique angle, which hampers interpretation.
- Standard presentation with the epidermis oriented at the top is recommended for clarity and consistency.

In summary, while the manuscript contains potentially valuable content, it lacks structural clarity and thematic coherence. Major revision is needed, including clearer connections among the different topics and clarification of clinical background details.

Author Response

Editor-in-Chief

Reviewers

We very much appreciate having the opportunity of resubmitting our manuscript entitled “Erythrodermic Psoriasis in the Context of Emerging Triggers: Insights into Dupilumab-Associated and COVID-19-Induced Psoriatic Disease”.We first wish to thank all the Reviewers for their very positive and encouraging comments on our manuscript, and the many useful suggestions for its improvement. We have seriously taken all these concerns into consideration and appropriate changes have been made in the revised manuscript.

Reviewer 1

Overall Assessment

This manuscript reports a case of dupilumab-induced erythrodermic psoriasis, provides a review of erythrodermic psoriasis triggered by COVID-19, and presents histopathological findings from a mouse model of long COVID. While each topic may be of interest individually, the rationale for combining these distinct subjects into a single manuscript is unclear. The lack of coherence among the topics makes the manuscript confusing. The authors should clearly explain why these three topics were integrated into one paper.

We have now added paragraphs to the revised manuscript explaining the connection. Furthermore, we created a schematic of the immunological mechanism to make it more comprehensive.

Concerning Dupilumab and Erythrodermic Psoriasis

The case of erythrodermic psoriasis following dupilumab administration raises several clinical concerns. In the reviewer's country, dupilumab is approved for atopic dermatitis, asthma, prurigo nodularis, and chronic spontaneous urticaria, but not for psoriasis.
- Was the patient treated off-label for psoriasis, or did they have a coexisting condition such as atopic dermatitis or asthma that justified the use of dupilumab?
- Is dupilumab approved for psoriasis in the United States or in the authors’ country? This point should be clarified.

  • Thank you for bringing this important point, we added information explaining what happened. It was off label trial as per his dermatologist to improve his condition. The dermatologist informed him about dupilumab, which has been used in some patient with psoriasis and shows some benefits. The patient wanted to try it. Interestingly, our case is the third case of developing erythrodermic psoriasis post dupilumab

Omission of COVID-19 Vaccination-Related Exacerbation

There are numerous reports of psoriasis flares and erythrodermic psoriasis following COVID-19 vaccination, not only infection.
- The manuscript does not address vaccination at all, which is a significant oversight.
- Did the patient in the reported case receive a COVID-19 vaccine? If so, the timing and type of vaccination should be discussed in relation to the onset of symptoms.

  • Thank you for this valuable suggestion. Now we added a section 4 entitle “Psoriasis/Erythrodermic psoriasis and COVID vaccines” in the revised m/s. Also we included a table for all the cases linked between covid-19 vaccines and psoriasis.
  • Our patient did not receive COVID, flu, or pneumonia vaccines before.

Unclear Purpose of Long COVID Mouse Model Pathology

The inclusion of histopathological findings from a long COVID mouse model is confusing.
- The relevance of these findings to erythrodermic psoriasis or the case presented is not explained.
- Moreover, the pathological features shown do not resemble those of erythrodermic psoriasis. If there is any similarity to human long COVID tissue, the authors should elaborate.

  • Thank you for your thoughtful feedback. We agree that understanding the mechanistic overlap among dupilumab-associated psoriasis flares, SARS-CoV-2 infection, and COVID-19 vaccination is crucial. While current literature does not provide definitive mechanistic studies linking these triggers to psoriasis exacerbation despite several cases reported among them, our observations suggest a potential shared immunopathogenic pathway. Specifically, pro-inflammatory mediators such as IL-1, IL-6, IL-17, TNF-α, Toll-like receptors (TLRs), NF-κB, and activated T lymphocytes are central to the pathophysiology of psoriasis and appear to be commonly involved in all three scenarios. Additionally, we noted histopathological similarities between dupilumab-induced psoriasis flares and those seen in coronavirus infection in our animal model, particularly collagen deposition and inflammatory cell infiltration, etc. Therefore, the cases of psoriasis can be surge in individuals infected with COVID-19 in the near future. Given that been said, we added all the necessary information in the manuscript.

Technical Issue with Histological Image Presentation

- The histological images are presented at an oblique angle, which hampers interpretation.
- Standard presentation with the epidermis oriented at the top is recommended for clarity and consistency.

  • Kindly note that the presented image represents the actual sectioning and staining of the tissue. Given the complexity of the process, achieving a flawless cut and perfect fixation can be challenging. However, these minor imperfections do not compromise the clarity or integrity of the key histopathological findings demonstrated in the H\&E staining. The destruction of the dermal architectures is clearly observed in both images.

    In summary, while the manuscript contains potentially valuable content, it lacks structural clarity and thematic coherence. Major revision is needed, including clearer connections among the different topics and clarification of clinical background details.

    • Thank you, we now have revised the manuscript based on the suggestion.

Sincerely,

Reviewer 2 Report

Comments and Suggestions for Authors

I read with great interest the manuscript. However, I have some comments:

-The abstract needs a better structure because it is a bit disorganized and can be confusing for the reader.

-line 38 “COVID-19”

-Fig 1. Please write an analytical caption

-In lines 133-152 you can summarize the biopsy finding and the conclusion etc. you don’t have to mention yellow star, black arrow etc on the main text. You can describe the histopathological photos analytically in the caption for each figure 2-4

-Fig 2. some of the labels appear to overlap, making the text difficult, even impossible to read. Could you please adjust the formatting?

-Table 1. Please add the number of references next to the authors. There is no need to write the title of the papers. Also it needs a better caption. You can divide the lines in sex, age, type of psoriasis, concomitant conditions, previous medical history of psoriasis or new onset? , time after covid-19 infection, drugs administered for Covid-19 etc to increase the clarity and readability of the table.

-lines 276-278: I think you mean Fig 5 instead of fig 4? Please fix it.

-In lines 263-273: as mentioned before, you can summarize the biopsy finding and the conclusion etc. you don’t have to mention stars, black/red arrows etc on the main text. You can describe the histopathological photos analytically in the caption for figure 5.

-as an idea, you could develop a schematic representation of the underlying immunological mechanism.

Overall, this article is a solid effort. With some revisions, it can be greatly enhanced.

Author Response

Editor-in-Chief

Reviewers

We very much appreciate having the opportunity of resubmitting our manuscript entitled “Erythrodermic Psoriasis in the Context of Emerging Triggers: Insights into Dupilumab-Associated and COVID-19-Induced Psoriatic Disease”.We first wish to thank all the Reviewers for their very positive and encouraging comments on our m/s, and the many useful suggestions for its improvement. We have seriously taken all these concerns into consideration and appropriate changes have been made in the revised m/s.

Reviewer 2

I read with great interest the manuscript. However, I have some comments:

- Thank you so much, we truly appreciated your comments and suggestions

-The abstract needs a better structure because it is a bit disorganized and can be confusing for the reader.

- We have revised the abstract and improved it significantly.

-line 38 “COVID-19”

- Sorry for the typo, we now corrected it

-Fig 1. Please write an analytical caption

-Thank you for the suggestion, we now modified as per the recommendation

-In lines 133-152 you can summarize the biopsy finding and the conclusion etc. you don’t have to mention yellow star, black arrow etc on the main text. You can describe the histopathological photos analytically in the caption for each figure 2-4

- We have now modified the text as per your suggestion. Thank you for this valuable comment

-Fig 2. some of the labels appear to overlap, making the text difficult, even impossible to read. Could you please adjust the formatting?

-We now changed the format and the legend.

-Table 1. Please add the number of references next to the authors. There is no need to write the title of the papers. Also it needs a better caption. You can divide the lines in sex, age, type of psoriasis, concomitant conditions, previous medical history of psoriasis or new onset? , time after covid-19 infection, drugs administered for Covid-19 etc to increase the clarity and readability of the table.

- We now changed based on the recommendation. Also, we added Table 2 with the same design.

-lines 276-278: I think you mean Fig 5 instead of fig 4? Please fix it.

- We apologize for inadvertent error, we fixed it.

-In lines 263-273: as mentioned before, you can summarize the biopsy finding and the conclusion etc. you don’t have to mention stars, black/red arrows etc on the main text. You can describe the histopathological photos analytically in the caption for figure 5.

- We have revised the text based on the recommendation.

-as an idea, you could develop a schematic representation of the underlying immunological mechanism.

- Done, Figure 6 illustrated the schematic explanation. Great suggestion, much appreciated. 

Overall, this article is a solid effort. With some revisions, it can be greatly enhanced.

-Thank you so much

Sincerely, 

Round 2

Reviewer 1 Report

Comments and Suggestions for Authors

The authors have responded appropriately and carefully to my concerns in the revised manuscript, and overall, the content has been improved. However, I believe there are important misinterpretations in the analysis of Figures 2 and 4. Before final acceptance, I would recommend addressing the following specific points:

  1. Please clearly state in the manuscript that informed consent was obtained from the patient who received dupilumab.
  2. Regarding Figure 2:
    (a) The yellow star appears to indicate the epidermis, not the dermis. Please consider revising this.
    (b) The structures indicated by the green arrows exhibit features of vascular endothelial cells. Therefore, they are more likely to be capillaries rather than interstitial blisters.
  3. Regarding Figure 4:
     The large central structure appears to be smooth muscle rather than a cluster of inflammatory cells. Please verify this and revise as necessary.

Author Response

The authors have responded appropriately and carefully to my concerns in the revised manuscript, and overall, the content has been improved. However, I believe there are important misinterpretations in the analysis of Figures 2 and 4. Before final acceptance, I would recommend addressing the following specific points:

Thank you very much for all the suggestions you provided to help improve our manuscript.

1. Please clearly state in the manuscript that informed consent was obtained from the patient who received dupilumab.

  • We appreciated your suggestion and revised the text accordingly

2. Regarding Figure 2:
(a) The yellow star appears to indicate the epidermis, not the dermis. Please consider revising this.

  • We apologize for the inadvertent error; the legend has now been modified

(b) The structures indicated by the green arrows exhibit features of vascular endothelial cells. Therefore, they are more likely to be capillaries rather than interstitial blisters.

  • We apologize for the inadvertent error; the legend has now been modified

3. Regarding Figure 4:
 The large central structure appears to be smooth muscle rather than a cluster of inflammatory cells. Please verify this and revise as necessary.

  • We apologize for the inadvertent error; the legend has now been modified

Thank you again for your time

Sincerely

Reviewer 2 Report

Comments and Suggestions for Authors

The authors have taken my suggestions into account and the manuscript has improved as a result.

*A small typo: in lines 245 +247, I believe it must be Figure 5 instead of Fig 4

Author Response

Thank you very much for all the suggestions you provided to help improve our manuscript.

The authors have taken my suggestions into account and the manuscript has improved as a result.

*A small typo: in lines 245 +247, I believe it must be Figure 5 instead of Fig 4

Thank you for the time and effort devoted to improving the manuscript. We apologize for the inadvertent error and have now revised the text accordingly.

Sincerely